biophysics

molecular dynamics, asymmetric bilayer, phosphoinositides

**Author for correspondence:**
Ravi Radhakrishnan
e-mail: rradhak@seas.upenn.edu

†These authors contributed equally to this work.

# Divalent cations bind to phosphoinositides to induce ion and isomer specific propensities for nano-cluster initiation in bilayer membranes

Ryan P. Bradley[1],[†], David R. Slochower[2],[†],
Paul A. Janmey[2,3] and Ravi Radhakrishnan[1,2,4]

[1]Department of Chemical and Biomolecular Engineering, [2]Graduate Group in Biochemistry and Molecular Biophysics, [3]Department of Physiology, and [4]Department of Bioengineering, University of Pennsylvania, Philadelphia, PA 19104, USA

RR, 0000-0003-0686-2851

We report all-atom molecular dynamics simulations of asymmetric bilayers containing phosphoinositides in the presence of monovalent and divalent cations. We have characterized the molecular mechanism by which these divalent cations interact with phosphoinositides. $Ca^{2+}$ desolvates more readily, consistent with single-molecule calculations, and forms a network of ionic-like bonds that serve as a 'molecular glue' that allows a single ion to coordinate with up to three phosphatidylinositol-(4,5)-bisphosphate (PI(4, 5)P$_2$) lipids. The phosphatidylinositol-(3,5)-bisphosphate isomer shows no such effect and neither does PI(4, 5)P$_2$ in the presence of $Mg^{2+}$. The resulting network of $Ca^{2+}$-mediated lipid-lipid bonds grows to span the entire simulation space and therefore has implications for the lateral distribution of phosophoinositides in the bilayer. We observe context-specific differences in lipid diffusion rates, lipid surface densities and bilayer structure. The molecular-scale delineation of ion-lipid arrangements reported here provides insight into similar nanocluster formation induced by peripheral proteins to regulate the formation of functional signalling complexes on the membrane.

†These authors contributed equally to this work.

# 1. Introduction

The interface between the intracellular and extracellular environment, defined by the boundary of the cell's plasma membrane, is the location of bidirectional outside-in and inside-out signalling responses. Such signalling pathways often involve a class of rare phospholipids known as polyphosphoinositides (PPIs) which can act as the starting point for second messengers and serve directly in transmitting signals. PPIs are among the most high-charge-density lipids on the cell membrane, their negative charge density originating through the deprotonation of two phosphomonoester groups that have pKa values in the physiological range of 6–8 and the negatively charged phosphodiester linker [1,2]. PPIs with two phosphate groups connected to the inositol ring—we focus on phosphatidylinositol-(4,5)-bisphosphate (PI(4, 5)P$_2$) and phosphatidylinositol-(3,5)-bisphosphate (PI(3, 5)P$_2$) in this study—can range from $-3e$ to $-5e$ depending on pH and the counterions present, whereas most mammalian lipids are neutral, zwitterionic, or carry a charge of $-1e$, see figure 1 [3].

Despite their low abundance in terms of the overall numbers, numerous cellular processes, including cytoskeletal assembly, are controlled by the level and spatial localization of PPIs on the cell membrane [4]. Among the PPIs, PI(4, 5)P$_2$, at about 1% mole per cent of all phospholipids, [5] is the most well studied. PI(4, 5)P$_2$ has been shown to act as a signalling beacon and a platform for microscale membrane clusters containing proteins, phospholipids, and cholesterol *in vivo* [6,7]. Hundreds of cytosolic proteins encompassing at least 10 different domains have been shown to bind PI(4, 5)P$_2$ either through non-specific electrostatic interactions or specific coordination of the phosphomonoester groups [8]. Furthermore, PPIs bind many cytoskeletal proteins and proteins that cause or sense membrane curvature. The shape of the cell membrane is controlled by changes in membrane-cytoskeletal linkages and forces generated by actin polymerization or cytoskeletal motors, all of which are sensitive to PPIs and their spatial organization [9–11]. Aside from their interactions with proteins, PPIs have been implicated in a number of diseases [12–14].

The positioning of phosphate groups on the inositol ring appears to bestow distinct biological roles to PI(4, 5)P$_2$ and PI(3, 5)P$_2$, despite their similar chemical structures. The significant charge of these molecules makes them not only sensitive to charged amino acids on protein domains, [8] but also to positive ions present in biological contexts, such as the monovalent ions Na$^+$ or K$^+$ and the divalent ions Ca$^{2+}$ and Mg$^{2+}$ [15–17]. The detection and visualization of PPIs *in vivo* has thus far relied on indirect perception through labelled proteins or chemical modification of PPIs with bulky dyes.

Owing to these limitations, one of the unresolved issues in understanding how phosphoinositides regulate cytoskeletal assembly and membrane curvature is how these scarce lipids affect the activity of many proteins [18]. Measurement of PI(4, 5)P$_2$ diffusion shows that most of the plasma membrane PI(4, 5)P$_2$ pool is bound or sequestered to some extent [19]. A major question is how PI(4, 5)P$_2$ distributes laterally within the plasma membrane [20,21]. Several recent studies show the relevance of nano-scale PI(4, 5)P$_2$ clusters to critical cellular functions [7,22–24]. PI(4, 5)P$_2$-dependent clustering of syntaxin 1 in the presynaptic plasma membrane is reduced when cholesterol is depleted [25], but PI(4, 5)P$_2$ clustering is not reduced when cortical actin assembly is disrupted with inhibitors, suggesting that PI(4, 5)P$_2$ clusters precede and do not result from actin nucleation. Strikingly, even in the absence of proteins, μM Ca$^{2+}$ induces PI(4, 5)P$_2$ clusters with a similar size distribution to those found *in vivo* [26]. In recent work, Wen *et al.* [27,28] reported the formation of multivalent cation-bridged PI(4, 5)P$_2$ clusters (approx. 8 nm) by using fluorescence self-quenching and Förster resonance energy transfer assays. Those experiments showed that concentrations as low as 0.05 mol% of PI(4, 5)P$_2$ cluster in the presence of divalent cations, with Ca$^{2+}$ being more efficient than Mg$^{2+}$ (i.e. clusters form at a lower ion concentration). However, the authors noted that they do not yet have a molecular model to explain the mechanism by which multivalent metal ions promote clustering [27].

In this work, we define how PPIs behave in model asymmetric membranes (i.e. multi-component lipid bilayers with a distinct inner leaflet), characterize their interactions with cations, and describe their propensity to initiate nanoscale clusters using all-atom molecular dynamics simulations. We note that owing to the limitations of large system sizes and computational cost, we do not aim to model cluster growth using atomistic models, but rather aim to delineate how the molecular context of the lipid-ion interactions can lead to conditions that initiate or seed clusters of phosphoinositides. Many modelling studies of biological membranes have used symmetric leaflet compositions and predominantly feature short-length acyl chains [10,11,29–31]. We modelled multiple membrane compositions to independently test several variables. Specifically, we varied the membrane composition, the presence or absence of cholesterol, phosphoinositide net charge and counterion species. We also used membranes with symmetric leaflets to compare with the asymmetric membranes. The effects of Na$^+$ and K$^+$ were contrasted with those of Ca$^{2+}$ and Mg$^{2+}$. We likewise compared the self-association and ion-binding properties of PI(3, 5)P$_2$ and PI(4, 5)P$_2$ isomers. We controlled for non-specific electrostatic attraction by

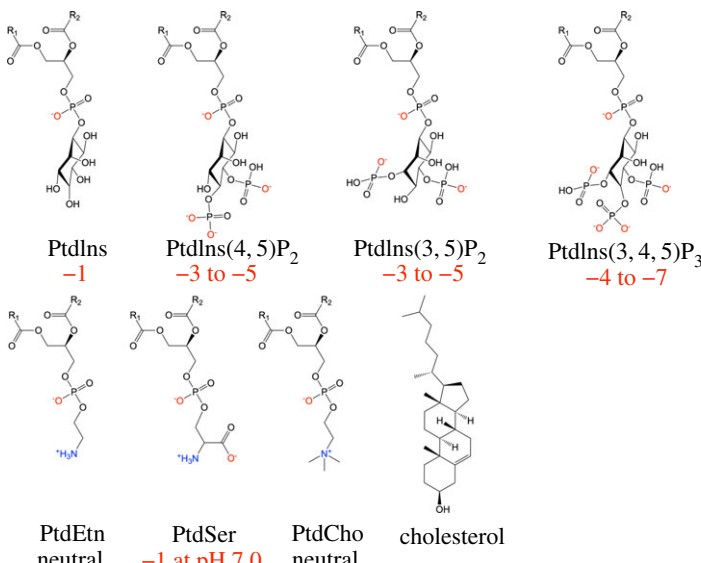

**Figure 1.** Chemical structures of the membrane constituents simulated in this work. The net charge is noted under each structure.

running identical simulations with the net charge of PI(4, 5)$P_2$ of $-3e$, $-4e$ and $-5e$. Simulation compositions are provided in the electronic supplementary material, table S1.

Our study proposes a model for context-specific cluster initiation observed in recent experimental studies [27]. Molecular dynamics simulations are capable of resolving the driving forces intrinsic to the biological function of PPIs which are typically beyond the scope of experimental techniques because they require combined knowledge of the lipid-lipid, lipid-ion and ion-water configurations. While our focus in this work is on ion-mediated PPI interactions, extension to protein binding and signalling are logical next steps [10,11]. The implications beyond the results stated are relevant to protein recruitment and subsequent control of cellular processes such as cytoskeletal rearrangements [11].

## 2. Methods

We employ atomistic molecular dynamics simulations in order to resolve lipid-lipid, lipid-ion, and ion-water binding in atomic detail. The physical properties of simulated membranes are sensitive to the force field parameters. We use the CHARMM36 (C36) lipid force that was optimized to correctly model the surface tension and area per lipid of choline- and ethanolamine-containing head groups with both saturated and unsaturated chains and also includes cholesterol [30,32,33]. Notably, parameters for PPIs exist within C36 and have been independently validated against other condensed phase measurements, such as lipid tilt, membrane electrostatic potential, and electron density [31,34]. We also use the NBFIX correction for ion interactions [35]. Among a comparison of lipid-specific force fields, C36 was found to most accurately describe lipid dynamics, with a slight bias towards lipid order relative to experiment, and overall good computational efficiency [36]. Although there can be overly favourable anion-cation pair interactions in many additive force fields, including C36 [37,38], a full reparameterization of divalent cations was not attempted here; nevertheless, we note that our results are qualitatively consistent with previous quantum mechanics/molecular mechanics calculations [39]. We used GROMACS 4.6.3 to simulate our model bilayers [40], with net charge and protonation states for PI(4, 5)$P_2$ and PI(3, 5)$P_2$ based on quantum calculations [41]. Longer (500 ns) simulations for two key systems, $Ca^{2+}$ versus $Mg^{2+}$, were performed using GROMACS 2018.2. We note, based on earlier works reported in the literature, that the timescale for ion binding and equilibration is 50–100 ns [42], while timescales for lipid mixing and rearrangements occur in the timescale of 500–1000 ns [43]. Of course, we cannot entirely rule out a scenario which could show a rearrangement with respect to ion binding at a longer time frame. However, we note that the results reported here reflect reproducibility of the observations across our replicates. Therefore, we expect our results to be valid for ion binding and equilibration and initiation of cluster formation but not fully capture growth of clusters.

We have investigated fifteen distinct compositions. Each bilayer has 400 molecules (i.e. lipids and cholesterol) in each leaflet. The physiologically composed bilayers have an asymmetric composition with 75% 1-palmitoyl-2-oleoyl-sn-glycero-3-phosphocholine (POPC) and 25% cholesterol in the outer

leaflet and 50% 1,2-dioleoyl-sn-glycero-3-phosphoethanolamine (DOPE), 25% cholesterol, 15% 1,2-dioleoyl-sn-glycero-3-phospho-L-serine (DOPS), and 10% PI, PI(4, 5)$P_2$, or PI(3, 5)$P_2$ in the inner leaflet. Our control systems included symmetric bilayers containing a 4 : 1 mixture of 1,2-dioleoyl-sn-glycero-3-phosphocholine (DOPC) and DOPS along with 10% PI(4, 5)$P_2$ or PI(3, 5)$P_2$ in both leaflets. A comprehensive list of simulation compositions can be found in the electronic supplementary material, table S1. We note that in preparing the asymmetric leaflets, we chose to focus on the compositions and balanced the number of lipids on each leaflet based on area per head group and the total leaflet area. These effects are in part mediated by our use of semi-isotropic pressure coupling to produce an approximately tensionless bilayer. In general, it is quite difficult to ensure the differential stress build-up across both the leaflets in an asymmetric simulation. Recent work [44] has indicated that area asymmetries below 5% have negligible effects on bilayer properties (e.g. thickness, diffusion, and order parameters). Our area mismatch is well below this 5% threshold.

A randomized grid of 400 molecular structures was assembled for each leaflet and composition and arranged with a regular 1 nm spacing. The lipids were fixed with mild position restraints of 500 kJ/(mol · $nm^2$) in the normal direction and then gently packed into a bilayer using a vacuum equilibration procedure in order to ensure that no lipids flipped to the opposite leaflet. A number of similar bilayer-construction tools are available [45], however, our procedure is closely aligned with the tools described in recent studies [46]. The resulting bilayer configurations were solvated with water and then one of four different cations ($Na^+$, $K^+$, $Mg^{2+}$ or $Ca^{2+}$) and $Cl^-$ ions at an ionic strength of 150 mM. As per prescription of Klauda *et al.* [30], we employed CHARMM 'special' water (TIPS3P) which includes Lennard–Jones interactions on water hydrogen atoms in order to prevent artefacts in lipid motion.

The asymmetric lipid compositions were chosen because they reflect the physiological composition of mammalian cell membranes [47,48]. As for soluble ion compositions, they are different inside and outside the cell, but they are much more similar than the lipid asymmetry in the bilayer. As for the physiological ion compositions, we note that the ion compositions of the combined monovalent ions ($Na^+$ plus $K^+$ combined) are roughly equal in the cytosol and the extracellular medium. The $Mg^{2+}$ compositions are also similar between the cytosol and the extracellular medium. Only the $Ca^{2+}$ concentrations are very different on either side of the cell mammalian membrane. But because the $Ca^{2+}$ concentration is higher on the extracellular side where the concentrations of highly charged polyanionic lipids like PI(4, 5)$P_2$ are negligible, the ion does not bind the lipids at this interface significantly. Therefore, the single bilayer set-up with periodic boundaries considered here will not influence the main conclusions. In particular, the electrical double layer is formed and stable adjacent to the inner leaflet. The total ionic strength is very similar, and the largest relevant difference is the much higher concentration of calcium ions outside the cell, an effect that would only strengthen our interpretation. The double bilayer set-up is too prohibitive on the basis of computational cost, but is a worthwhile exercise for future studies of the systems we have considered here. Our systems were equilibrated for at least 20 ns followed by production runs lasting 80–500 ns. These systems were maintained at a temperature of 310 K using the velocity-rescaling thermostat of Bussi *et al.* [49] with a coupling frequency of 0.5 $ps^{-1}$ and atmospheric pressure via the Parrinello–Rahman barostat with a coupling constant of 2.0 $ps^{-1}$. The LINCS algorithm constrained hydrogen bond distances allowing for a 2 fs timestep. Van der Waals forces were switched off smoothly from 0.8 to 1.2 nm, and electrostatics were computed according to the particle-mesh Ewald summation with a Fourier spacing of 0.16 nm.

We report a number of bilayer properties using a set of analysis codes written in Python which make use of several libraries based on Python's NumPy [50], namely MDAnalysis [51] for reading and selecting molecules and SciPy [52] for measuring lipid-ion-water distances under periodic boundary conditions using a *k*-dimensional tree [53]. The distances and angles were also used to identify associations including hydrogen bonds and salt bridges. Diffusion coefficients are computed from linear fits of the lateral mean-squared displacement after accounting for periodic boundary conditions and our bilayers have no net drift artefacts.

We quantify the lipid-lipid lateral distribution using two-dimensional meshes of the lipid centres of mass. We identify the observed numbers of pairs (two adjacent lipids) and triplets (three adjacent lipids) from a Delaunay triangulation generated using the quickhull algorithm [54] within SciPy with modifications to generate triangulations across periodic boundary conditions. We used the same algorithm to compute a Voronoi tesselation that allowed us to partition the leaflet area among each lipid species. This procedure contributed to lipid area calculations and a description of lipid association propensities discussed in the Results section. These methods are available in open-source format (via https://biophyscode.github.io/). Unless specified otherwise in this manuscript, PIP$_2$ refers to either PI(4, 5)$P_2$ or PI(3, 5)$P_2$.

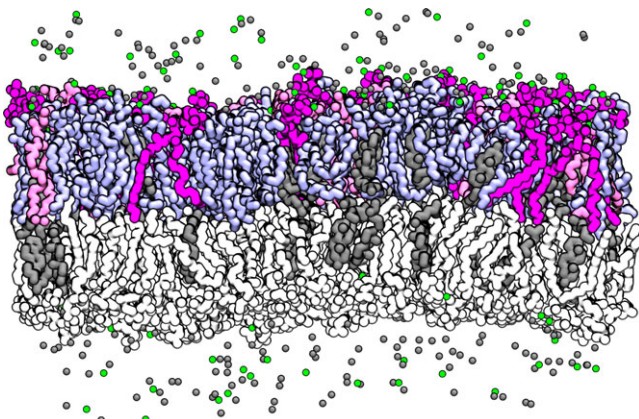

**Figure 2.** Simulation snapshot of a bilayer containing 800 lipids with PIP$_2$ (purple) found in the inner leaflet (top) along with DOPE (light blue), DOPS (pink) and cholesterol (dark grey). The outer leaflet (bottom) contains POPC (white) and cholesterol. Ca$^{2+}$ cations are shown in green and chloride ions are grey spheres.

All simulations were performed in replicates on several supercomputing platforms provided by XSEDE as well as on a local cluster. Simulations contained up to 360 000 atoms including water and counterions (see the electronic supplementary material, table S1) and achieved simulation speeds of up to 16 ns d$^{-1}$ on $6 \times 16$-processor platforms. The simulations ran for an aggregate production time over 2 µs. Convergence was assessed by comparing the radial distribution functions along the trajectories of different lengths, see the electronic supplementary material, figures S1 and S2.

## 3. Results

A typical simulation snapshot is provided in figure 2. All such snapshots, including the electronic supplementary material videos, have been rendered with visual molecular dynamics (VMD) [55]. An example simulation is depicted in the electronic supplementary material, video SV1.

### 3.1. Ion-lipid head group coordination drives ion-specific clustering of PIP$_2$

To test the central hypothesis that lipid-lipid, lipid-water and lipid-counterion interactions are essential determinants of the distribution of PPIs within a membrane, we first consider the formation of ion-lipid complexes. We classify the multivalent lipid-cation interactions by the number of lipids bound within a specific cutoff distance of each ion. We call these structures $N$-bridges where $N$ is the number of bound lipids. We select a cutoff distance that is equal to the radius of the first solvation shell in order to ensure that we only identify ions that are bound tightly enough to exclude water. These cut-offs were identified from the first local minima of the water-ion radial distribution functions (see the electronic supplementary material, figure S1), and are set to 3 Å for Na$^+$ and K$^+$, 2.3 Å for Mg$^{2+}$ and 2.6 Å for Ca$^{2+}$. Figure 3 shows the counts of 1-bridges, 2-bridges and 3-bridges for the 500 ns simulations.

The counts in figure 3 capture a striking result: namely that Ca$^{2+}$ forms tight, nearly ionic bonds with lipids—that is, those capable of excluding water—at a much higher rate and with a higher charge density than other ion species. The high $-4e$ charge on PIP$_2$ ensures that most of the ions in the simulation are drawn to the bilayer, forming an electric double layer. We refer to this phenomenon as *charging* [56]. Ca$^{2+}$ prefers to bind to PIP$_2$ more than any other lipid, as evidenced by the large solid bars, denoting ions bound to the 10% PIP$_2$ in the inner leaflet, compared to the hatched, grey bars which show bonds to the remaining 90% of the lipids in the bilayer. In contrast to Mg$^{2+}$, Na$^+$ and K$^+$ (see the electronic supplementary material, figure S3), we find that Ca$^{2+}$ forms many multivalent complexes that include at least one PIP$_2$. The ion binding results for additional systems, including those containing PI(3, 5)P$_2$, are shown in the electronic supplementary material, figure S3.

The distinction between Ca$^{2+}$ and Mg$^{2+}$ is also evident in the electronic supplementary material, video SV2 which shows that the lifetime of Mg$^{2+}$ binding to phosphate groups is shorter than Ca$^{2+}$, which rarely dissociates from the phosphate groups. This key difference is reflected in the number of Ca$^{2+}$ bound to at least three lipids (i.e. 3-bridges) including PIP$_2$, which grows progressively over the course of the simulation.

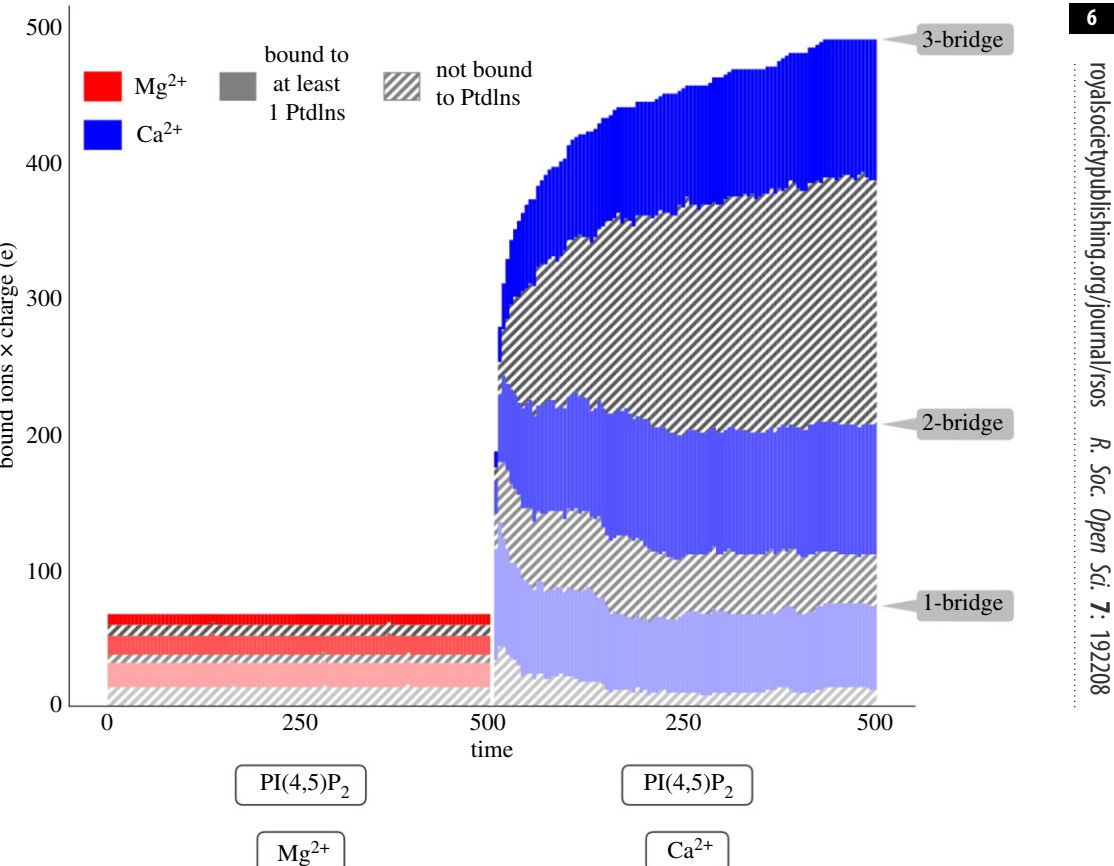

**Figure 3.** Counts of closely associated ions distinguished by the number of lipids which are also associated with these ions. We define bound ions according to the radius of the first solvation shell, which is 3.0 Å for $Na^+$ and $K^+$, 2.3 Å for $Mg^{2+}$, and 2.6 Å for $Ca^{2+}$. Colours correspond to one, two or three closely associated lipids, and are stacked in that order, i.e. from bottom to top as marked on the right. Solid, coloured bars show the number of ions associated with clusters with at least one $PIP_2$. In the case of the lightest colours, the bound ions are therefore bound to exactly one $PIP_2$ while the darkest colour includes ions associated with one or more $PIP_2$ along with other lipid species. The grey, hatched bars represent ions which are associated with other lipids, in this case DOPS or DOPE. We have excluded the outer leaflet lipids (POPC) in this calculation in order to highlight the inner leaflet. Bar colours correspond to the chemical identity of the ion (blue for $Ca^{2+}$, and red for $Mg^{2+}$). While $Ca^{2+}$ has a larger number of bound ions, they are also increasing over the duration of the simulation. Note, Ptdlns denotes phosphatidylinositol.

These rearrangements accompany a mild additional net increase in total bound $Ca^{2+}$ which is not seen in the other ions. This demonstrates that the bilayer is *charging* on the relatively slow, 500 ns timescale of the simulation, facilitated by a reorganization of the lipid configurations around the ions.

$Mg^{2+}$ is not able to generate 2-bridges or 3-bridges to any significant degree, nor does it accumulate on the membrane surface. It is interesting to note that $Na^+$ can also facilitate a smaller number of 3-bridges (see the electronic supplementary material, figure S3), however, these clusters rarely include $PIP_2$ and do not change appreciably with time. This suggests that $Na^+$ occupies the positions within pre-existing lipid configurations rather than affecting a reorganization that maximizes lipid contact with the ions. In membranes with identical leaflet compositions, we find that the $Na^+$ condensation is the highest when $PIP_2$ carries the highest charge (for example, when deprotonated compared to doubly protonated; see the electronic supplementary material, figure S3). This calculation shows almost no measurable differences between the $PI(4, 5)P_2$ and $PI(3, 5)P_2$ isoforms. $Na^+$ binds PI less than $PIP_2$, indicating that many of the bound cations occupy the exposed phosphate groups. In the presence of $PIP_2$, $K^+$ forms significantly fewer bonds than $Na^+$.

## 3.2. Lipid-ion-lipid bridging drives the growth of nanoclusters

To characterize the implications for the lateral distributions at length scales beyond the size of our simulations, we define a *cluster* as a set of lipids which are bound by at least one *N*-bridge and study

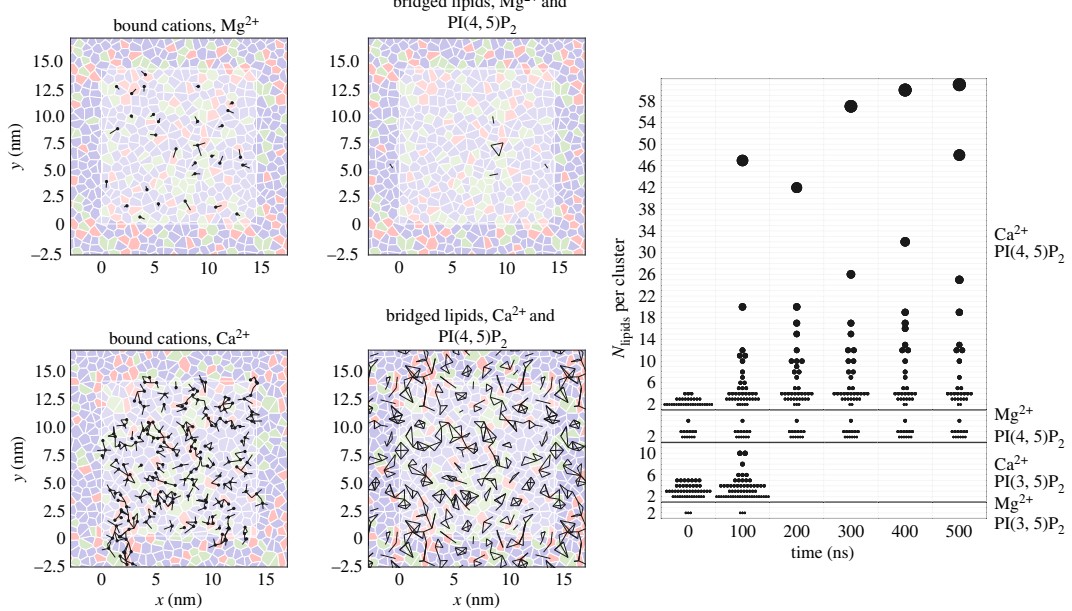

**Figure 4.** Diagram of observed $N$-bridges (left) and the resulting bond network (centre) alongside a quantification of the resulting clusters (right) over the duration of each simulation. Instantaneous snapshots (left and centre) were taken at the conclusion ($t =$ 100 ns) of two replicates. The left panels show an overhead snapshot of the Voronoi tesselation of lipids on the inner leaflet distinguished by colour ($PIP_2$ is red; DOPS is green; DOPE is purple). Black dots correspond to bound cations, with black lines indicating the tile (i.e. lipid) to which they are bound. Only the cation-lipid bonds for a single periodic image are shown. In the schematic in the centre panels, we reduce the lipid-cation-lipid bonds each to a single lipid-lipid edge drawn between tiles and repeated for periodic images. The right panel depicts histograms of the resulting lipid clusters according to the number of lipids in each cluster. Each symbol represents a cluster of lipids. The symbol area is proportional to the number of lipids in the cluster, which is also measured on the vertical axis. Only the unique combination of $PI(4, 5)P_2$ and $Ca^{2+}$ shows large and growing clusters.

the distribution of clusters over time. We derive the clusters from the $N$-bridges by reducing each lipid-cation-lipid bond to a lipid-lipid bond which forms an edge in the graph for that cluster. Figure 4 provides diagrams of both the $N$-bridges and the resulting clusters for systems containing either $Mg^{2+}$ or $Ca^{2+}$. We derive the clusters by collapsing each lipid-cation-lipid bond which contributes to an $N$-bridge into a single edge in a graph of associated lipids. The size of each graph ($N_{lipids}$) measures the cluster size, while the distribution of these clusters (see figure 4, right panel) quantifies the extent to which the lipid motion is hindered by the 'molecular glue' formed by the cations to form a larger collective lipid mass. The largest clusters of up to 60 lipids are only observed for the combination of $PI(4, 5)P_2$ and $Ca^{2+}$. Simulations containing an alternate isomer, for example $PI(3, 5)P_2$ or PI, or a different ion, notably $Mg^{2+}$, fail to generate these large and growing clusters.

The following sections characterize the atomic basis for the lipid-cation bonds and derive the effects of these bond networks on the collective lipid dynamics by studying their diffusion rates and surface densities. In contrast to these local, nano-scale measures of the molecular structure of the bilayer, however, the network of strong bonds quantified here is large enough to span the simulation box and therefore has implications for the lipid distributions at larger length scales.

In a recently published article, Han *et al.* [57] characterized $PIP_2$ clusters in a somewhat similar fashion, albeit in monolayers composed entirely of $PIP_2$. Our results build on their recommendation to extend this investigation to bilayers and bilayers with other lipid species. The Han *et al.* analysis proceeds from a different definition of an 'edge' in the lipid-association graph, namely the proximity between phosphate oxygen groups. Our definition of an 'edge' is a lipid-cation-lipid bond, that is, a desolvated cation shared between two lipids. Both definitions are valid ways to describe the concept of a cluster, however, we are unlikely to observe the edges defined by Han *et al.* in our simulations because we have a much lower $PIP_2$ concentration and direct $PIP_2$ interactions are statistically much less likely because there are many other lipids. This is the primary reason for the alternate definition of an 'edge' within a cluster employed in this work (see also the Discussion section).

## 3.3. Ion hydration mediates ion-lipid interactions to modulate ion-specific lipid clustering

The main result of transient $Mg^{2+}$ binding versus sustained $Ca^{2+}$ binding in figure 3 is primarily governed by the level of ion hydration. This is apparent in the electronic supplementary material, video SV2, where a single $PIP_2$ in a bilayer is selectively depicted to highlight the interplay between the lipid, ions and water. The persistent binding of $Ca^{2+}$ is consistent with $Ca^{2+}$ interacting much more strongly with $PIP_2$ by rearranging its hydration shell, while the short-lived binding of $Mg^{2+}$ is consistent with the almost full hydration shell around the ion as it interacts with $PIP_2$, thereby leading to a greater separation between the ion and $PIP_2$. Before we quantify these observations to demonstrate that this qualitative picture is borne out, we note that the observations of a stronger dehydrated $Ca^{2+} - PIP_2$ bond and a weaker water-mediated (i.e. non-dehydrated) $Mg^{2+} - PIP_2$ bond agrees with single-molecule free energy calculations reported previously [41].

To examine the differences in ion-lipid-water complexes, we quantify the number of waters in the first hydration shell of these ions as they approach the lipids. We count the waters in the solvation shell according to equation (3.1):

$$\left. \begin{array}{l} N_{ions}(r, t) = \sum_{c,l} [|r - d_{c,l}(t)| \leq \epsilon] \\ N_{waters}(r, t) = \prod_{c,l} [|r - d_{c,l}(t)| \leq \epsilon] \\ \qquad \cdot \sum_{w} [d_{c,w}(t) \leq d_s] N_{ions}^{-1} \end{array} \right\}. \tag{3.1}$$

In this equation, the indices $c$, $l$, $w$ refer to cations, lipids, and water molecules, respectively; $\epsilon = 0.05$ Å sets the observation windows for the cation-lipid distances; $d_{i,j}(t) = \min_{m \in i} \min_{n \in j} r_m(t) - r_n(t)$ represents the minimum distance between any two atoms $m$, $n$ in a pair $i$, $j$ of molecules; $t$ is time; and $d_s$ refers to the radius of the first solvation shell. We discretize minimum distance between atoms ($r_{ij}$) into bins of size 0.05 Å. Note that the bracket operators yield Boolean values used to count each cation-lipid and cation-water distance in the correct distance bin. Equation (3.1) thereby quantifies the cations bound to lipids (at a distance $d_{c,l}$) and the waters bound to the cations (at a distance $d_{c,w}$); see snapshots in figure 5.

Figure 6 depicts the result, namely the distribution for $N_{waters}(r,t)$ across a range of cation-lipid distances along with the number of ions ($N_{ions}$) found at these distances. Although $Mg^{2+}$ forms a bond with $PIP_2$ at a closer distance than $Ca^{2+}$, there are many more $Ca^{2+}$ ions bound between 1.75 – 2 Å. $Ca^{2+}$ density in this zone is more than fourfold as high as $Mg^{2+}$, a result which is also reflected in the charging curves found in figure 3. $Mg^{2+}$ has a nearly equal preference for making either an ionic bond (approx. 2 Å) or occupying the region between 3 and 5 Å.

We find significant differences in hydration between $Ca^{2+}$ and $Mg^{2+}$. First, $Ca^{2+}$ more easily loses water, retaining only 2–4 waters at 2 Å. The strength of the $Ca^{2+} - $ lipid bond leaves comparatively few ions distant from any lipids, where they are coordinated by 4–6 waters, compared to 5–7 waters for $Mg^{2+}$. These findings suggest that both ions lose 1–2 waters upon forming an ionic bond with phosphate oxygen atoms, beginning around 3 Å from the lipid. We find almost no difference between simulations containing PI(4, 5)$P_2$ and PI(3, 5)$P_2$ in the histogram of ion counts and the hydration of each ion species (see the contrast in the electronic supplementary material, figure S4).

Snapshots of the lipid-ion-water configurations for $Mg^{2+}$ and $Ca^{2+}$ are shown in figure 5 and call attention to the less-structured and less-hydrated nature of $Ca^{2+} - $ PI(4, 5)$P_2$ binding. One can see that $Mg^{2+}$ ions are almost always buffered by water molecules while $Ca^{2+}$ bonds directly with the lipids.

The ion-water radial distribution functions (see the electronic supplementary material, figure S1) show that water adopts a different structure around $Mg^{2+}$ versus $Ca^{2+}$ at both near (less than or equal to 2.2 Å) and far (greater than or equal to 4.6 Å) distances from the lipids. Waters in the first and second solvation shell are closer to $Mg^{2+}$ than $Ca^{2+}$ in good agreement with an earlier report [58]. In contrast to $Mg^{2+}$, when $Ca^{2+}$ is close to lipids (electronic supplementary material, figure S1, bold blue line), the density of water in its second hydration shell drops 25% relative to when it is far (light blue line).

The fact that $Ca^{2+}$ forms tighter bonds with $PIP_2$, and has fewer waters in its first and second solvation shells when it does so, compared to $Mg^{2+}$, suggests that lipid-lipid interactions are context-specific, modulated by the ion species along with the particular structure of its hydration shell.

## 3.4. Inter-lipid hydrogen bonding

The charging curves in main text figure 3 quantify the number of closely bound cations, the hydration of those ions is characterized in figure 6 and the electronic supplementary material, figure S4, and their lipid-binding sites are characterized in figure 7. In this section, we analyse the effect of ion binding on the native

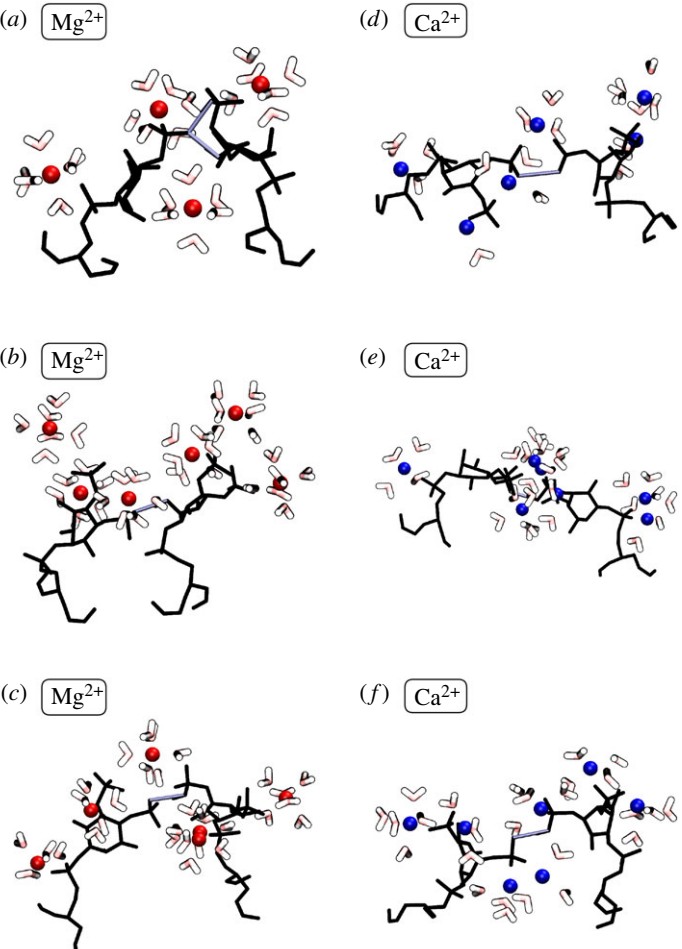

**Figure 5.** Snapshots of lipid-cation-lipid bonds. Each panel shows an instantaneous diagram of two PIP$_2$ (black), their associated cations (Mg$^{2+}$ in red and Ca$^{2+}$ in blue), along with associated water molecules in the first solvation shell (red for oxygen and white for hydrogen, rendered as a tube). Hydrogen bonds between lipids are depicted with a purple tube. We find that bonds between PIP$_2$ and Mg$^{2+}$ (a–c) show more water in the solvation shell than bonds between PIP$_2$ and Ca$^{2+}$ (d–f).

hydrogen-bonding network formed between the lipids, irrespective of the mediating cations. These lipid-lipid hydrogen bonds represent an alternate mechanism by which lipids self-associate which could either supplement or compete with bonds mediated by cations. Electronic supplementary material, figure S5 depicts lipids directly engaged in hydrogen bonds along with the flanking ions. Hydrogen bonds are defined by a donor-acceptor distance of 3.4 Å with a donor-hydrogen-acceptor angle greater than 160°.

In the electronic supplementary material, figure S5, it is clear that PIP$_2$ readily bonds with DOPE, but otherwise has mostly opportunistic bonding with other partners, including cholesterol. Overall, however, the hydrogen-bonding network is not strongly influenced by the identity of the cation. The first conclusion to draw is that the strong network of 2- and 3-bridges which we have described in the preceding sections does not compete with or interrupt the formation of hydrogen bonds. We find that PIP$_2$ forms one hydrogen bond on average 100% of the time with DOPE, compared to roughly 30% on average with DOPS. Bonds between two PIP$_2$ or bonds between a PIP$_2$ and cholesterol are comparatively rare, representing 10% of the duration of the simulation time. Hence salt bridges, and not hydrogen bonds, appear to dominate interactions between PIP$_2$, emphasizing the importance of their headgroups. This is further underscored by the comparison between PIP$_2$ and PI; the latter forms a hydrogen bond only 60% of the time with DOPE.

Electronic supplementary material, figures S6 and S7 show a complete tabulation of normalized salt bridges and hydrogen bonds. It is important to note that the salt bridge calculation captures the $N$-bridges described in the main text because it includes mediating cations, while the hydrogen-bonding patterns are measured between lipids, independent of the cations. The electronic supplementary material, figures S8 and S9 describe the lipid atoms which participate in these bonds.

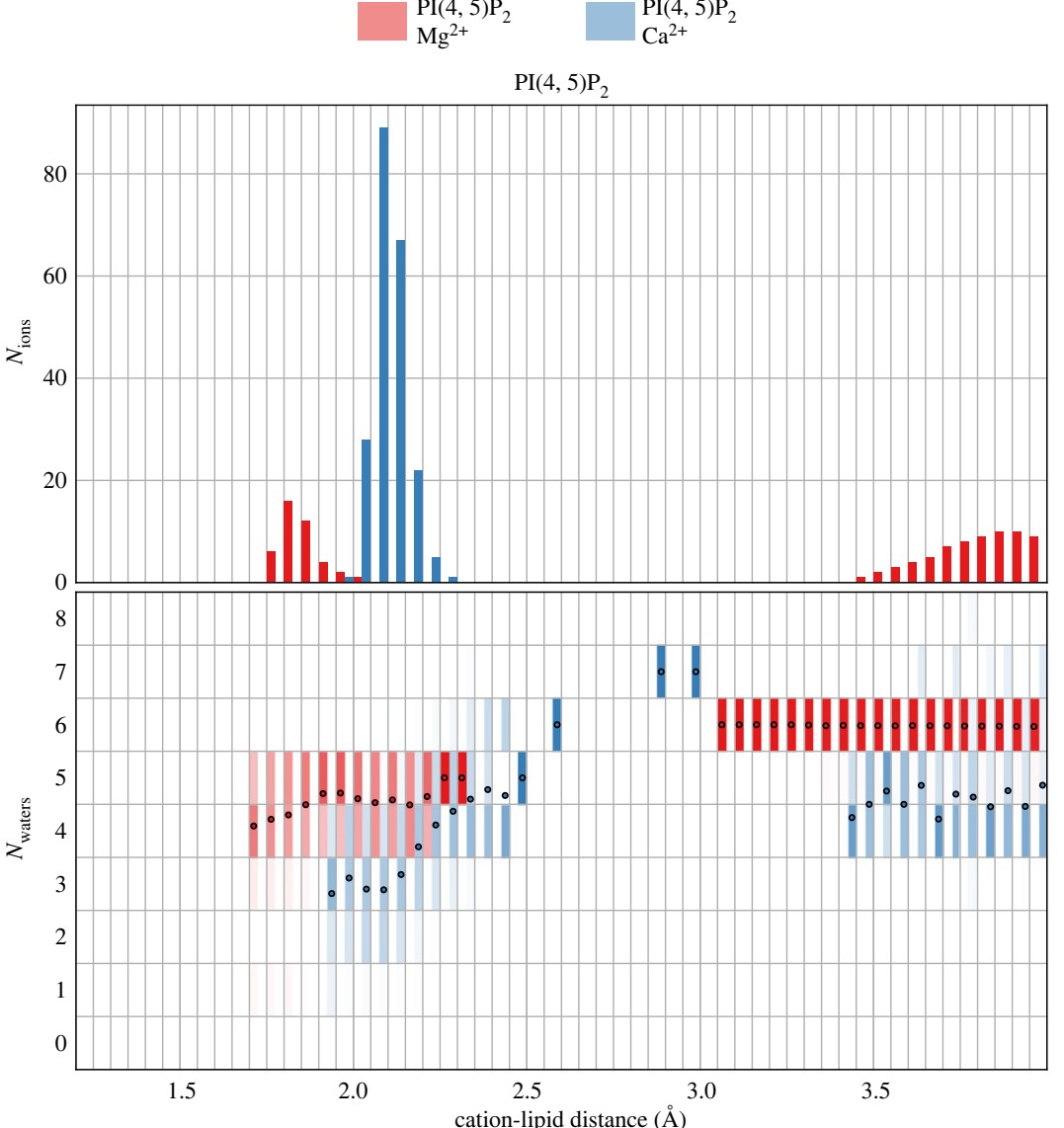

**Figure 6.** Number of bound ions $N_{ions}$ (top) as a function of minimum lipid-ion distance ($d_{cl}$) along with the corresponding number of water molecules bound to each ion $N_{waters}$ (bottom). In the bottom panel, the distribution of the number of waters at each cation-lipid distance is reflected in the colour intensity in each (integer) bin on the vertical axis, while a black dot indicates the mean number of hydrated waters at that distance. The colour intensity is normalized for each bin, so even if there are comparatively few observations of ions $N_{ions}$ at that distance (for example, between 2.5 and 3.3 Å), the preferred number of waters in the first shell is still shown. This scheme highlights some rare events in which $Ca^{2+}$ acquires up to seven waters at an intermediate distance (2.8 Å) from the lipids. Both $Mg^{2+}$ (red) and $Ca^{2+}$ are plotted together for comparison.

## 3.5. Which atoms facilitate lipid-ion-lipid bridging?

The quantification of ion hydration shells in the previous section shows that each ion interacts differently with water, and that $Ca^{2+}$ is more likely to have waters in its first hydration shell replaced by phosphate oxygens when it is close to the lipids. In this section, we identify the specific parts of each lipid that form a lipid-cation-lipid salt bridge in order to isolate the isomer- and ion-specific differences in these interactions. Figure 7 shows a heat map of the atoms that participate in these salt bridges. Note that salt bridges are mediated by cations and hence form the basis for the $N$-bridges.

We find that $K^+$ salt bridging occurs almost exclusively on the 4−phosphate group (e.g. atoms `OP42`, `OP43` and `OP44` in figure 7, named according to the `C36` [30] convention). PI forms comparatively few salt bridges with other PI, and the same is true for all lipids when $Mg^{2+}$ is present (see the electronic supplementary material, figure S13). Interestingly, PI(4, 5)P$_2$ is more likely to form a $Ca^{2+}$-mediated

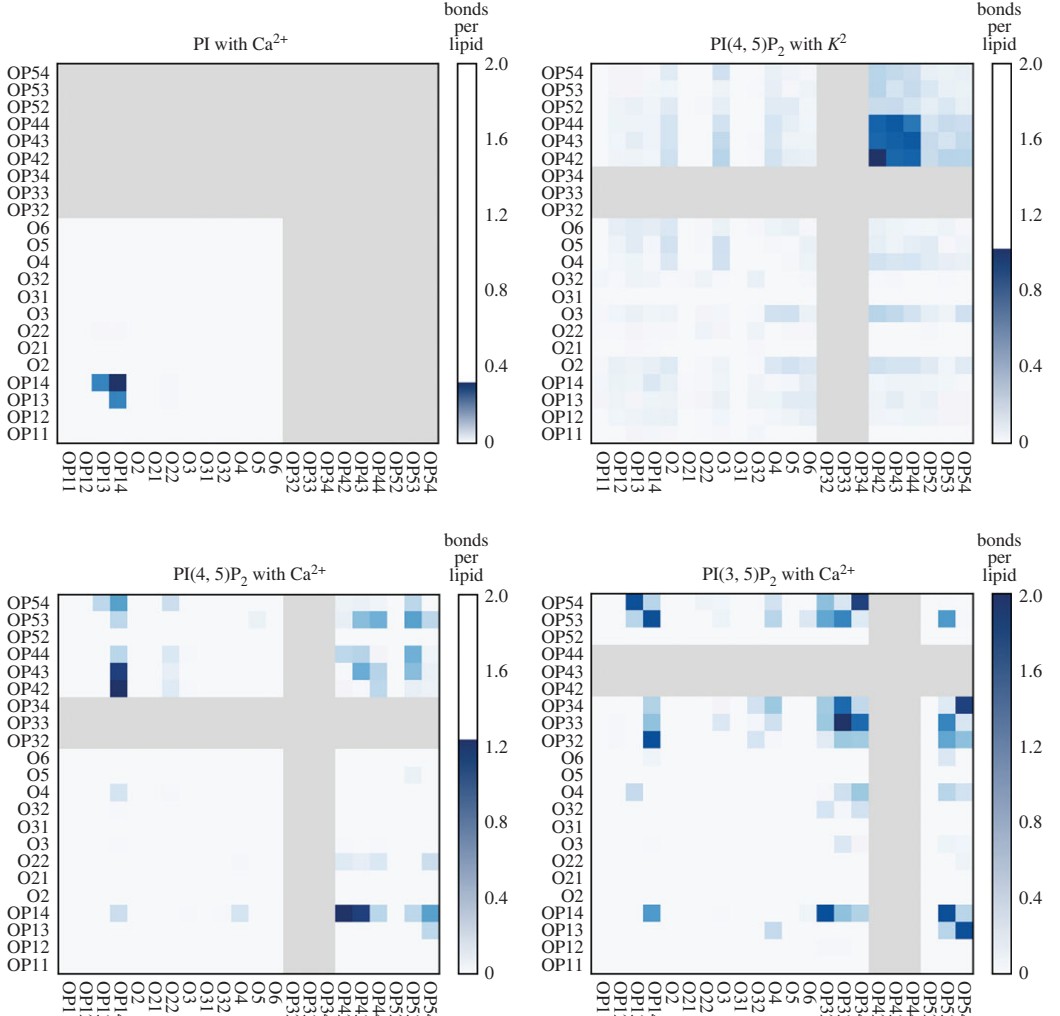

**Figure 7.** Salt bridges between two PIP$_2$ or (PI, upper left) for selected simulations. Each heat map shows the number of lipid-cation-lipid salt bridges (between heavy atoms within 3.4 Å) for each pair of valid atoms. Each axis uses the same list of atom names; atoms are greyed out if they are not members of that molecule. The heat maps are normalized to show the number of bridges per lipid. Each tile is normalized to a different maximum number of bridges indicated by the colour bar. Counts are provided in the electronic supplementary material: of atom-specific hydrogen bonds (electronic supplementary material, figure S8) and salt bridges (electronic supplementary material, figure S9). Atom names follow the CHARMM36 convention in which the OP4 prefix indicates the 4-phosphate oxygens. Our PIP2 are protonated at the OP52 position where there is a corresponding absence of bridges (lower intensity above). The prefix OP1 indicates the diester phosphate.

salt bridge between the diester phosphate on one lipid and the 4– and 5-phosphate groups on a different lipid, forming an asymmetric bond between lipids. By contrast, PI(3, 5)P$_2$ forms bonds between the 3- and 5-phosphate groups of different lipids, and also has slightly more overall salt bridges than Ca$^{2+}$ and PI(4, 5)P$_2$.

## 3.6. Effect of ion-lipid molecular association on macroscopic properties

Thus far, we have provided evidence that the strength and multivalency of Ca$^{2+}$ – PIP$_2$ bonds are qualitatively different from PIP$_2$ bonds formed with other ions, even with the same charge, namely Mg$^{2+}$. These bond networks form clusters which can span the simulation box. Here, we consider the possible implications of this large network on the behaviour of many lipids acting in concert. The features of the lipid-ion associations have a measurable effect on the collective material properties of the bilayer including the lipid diffusion rates and the lateral area of the leaflets containing PIP$_2$ molecules. While these properties are averaged over measurements of individual lipids, they are emblems of the larger structure of the bilayer.

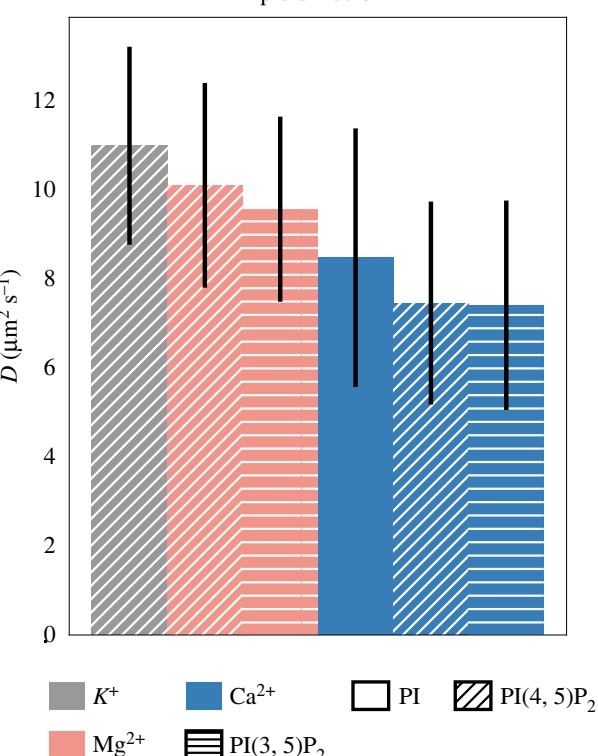

**Figure 8.** Diffusion rates according to lipid and cation identity. Lipid diffusion rates for PPI in our simulations. Error bars describe the standard deviation of the diffusion constant over the 40 PIP$_2$ molecules in each simulation. Simulations with K$^+$ and Na$^+$ have similar diffusion whereas Ca$^{2+}$ appears to reduce the diffusion rate slightly. See the electronic supplementary material, figure S10 for diffusion rates for the remaining simulations.

### 3.6.1. Diffusion coefficients

The strong lipid-lipid association owing to Ca$^{2+}$ creates a larger effective mass for diffusing lipids connected by ions. In particular, we hypothesize that these strong bonds, and specifically 3-bridges, make the effective diffusive unit larger in proportion to the number of additional lipids that are associated with the PIP$_2$ molecule.

Figure 8 shows the lipid diffusion rates for the asymmetric bilayers. We find that Ca$^{2+}$ slows the diffusion of all the lipids, in qualitative agreement with experiments [59]. PIP$_2$ diffuses the slowest of all lipids in each simulation, while PI diffuses about as fast as DOPS. Differences in lipid diffusion appear to be more sensitive to the ion species than the presence of cholesterol (see the electronic supplementary material, figure S10 for diffusion rates in the cholesterol-free, symmetric bilayers). We note that diffusion rates in finite simulations are subject to a number of measurement errors, most notably owing to the periodic nature of the simulation [60]. The diffusion rates reported here may be underestimating the true diffusion rates observed in a much larger simulation.

### 3.6.2. Bilayer area

The area per lipid is an important biophysical quantity because it encodes information about both the density and rigidity of lipid bilayers. It can be measured by experiments [30,61] and is one of the crucial quantities of comparison between molecular models and experiments. We have computed the lipid areas in two ways: first, using the three-dimensional (3D), rugged surface of each leaflet and secondly, by projecting the lipid centres of mass onto a two-dimensional (2D) surface, which controls for the excess out-of-plane area contribution of the PIP$_2$ headgroup. The 3D areas are depicted in figure 9 (see the electronic supplementary material, figure S11 for the 2D areas).

The projected lipid areas provide evidence that Ca$^{2+}$ condenses the total 2D area of the bilayer (see the electronic supplementary material, figure S11). We observe a reduction of an average of ~1.5 Å$^2$ per lipid in the simulations with Ca$^{2+}$ compared to the other conditions. Having constructed these bilayers with equal numbers of lipids in both leaflets, we find that the inner leaflet carries a larger 3D area across all

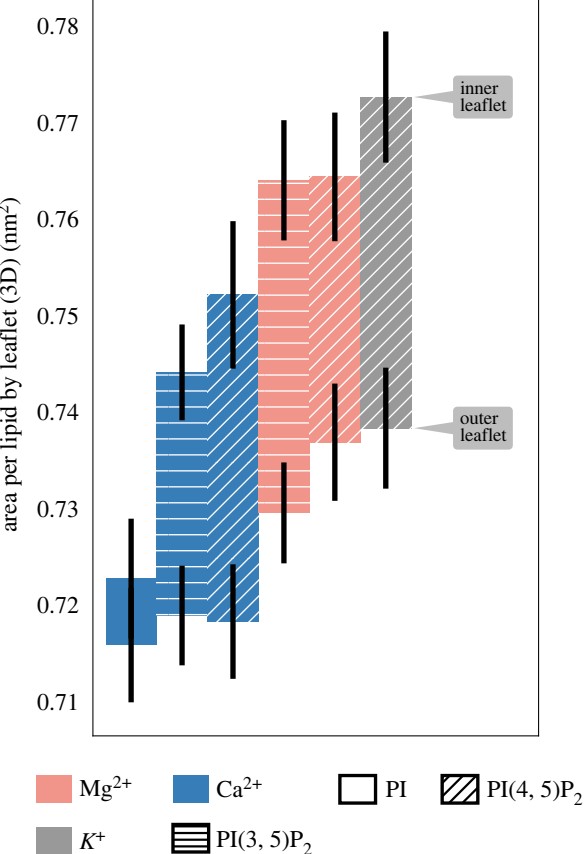

**Figure 9.** 3D leaflet areas for asymmetric bilayers. Leaflet areas are computed from the Delaunay mesh between lipid centres of mass under periodic boundary conditions. The top and bottom of each bar indicates the mean inner and outer leaflet area, respectively. We find that the PI simulation has statistically similar leaflet areas, while the $PIP_2$ simulations, which contain the same total number of lipids and cholesterol molecules, have an additional inner leaflet area.

systems which contain $PIP_2$ (figure 9, right panel), although the differential between leaflets varies slightly by cation. This extra inner-leaflet area is notably absent when PI is present, suggesting that the extra area is created by the additional phosphate groups carried by $PIP_2$. Even when these phosphate groups are absent, as is the case for PI, the $Ca^{2+}$ causes the bilayer to contract compared to the other conditions. The additional leaflet area produced by PPI species is underscored by the area distributions across individual lipids shown in the electronic supplementary material, figure S12. We also find that the area compressibility is somewhat elevated when $Ca^{2+}$ is present (see the electronic supplementary material, figure S13). The simultaneous 2D contraction along with extra 3D area suggests that PPIs under these conditions have a larger intrinsic curvature, which may affect their lateral distribution, and that their situation in the bilayer changes such that they 'bulge' more into solvent, providing greater access to the cations. We note that the observed effects on bilayer area condensation are consistent with experimental observations of area reduction in monolayer experiments reported by Levental *et al.* [62]. However, the magnitudes of such effects observed in our simulations (0–5% reduction in area) are smaller than those in the monolayer experiments (0–40%). While we cannot rule out force-field artefacts as cause for this deviation, it is reasonable that the deviations are attributable to monolayer versus bilayer and differences in lateral pressures between the simulations and the experiments.

## 3.7. Ion-lipid context mediates lipid association propensities

The simulations presented in this work span a timescale that is long enough to capture the motion of cations and corresponding changes in lipid structure which generate N-bridges and the bilayer-spanning bond networks depicted in figure 4. While this timescale is too short to observe long-range lipid diffusion, we can infer lipid association preferences by comparing the spatial arrangement of lipids between conditions.

We quantify the lipid-lipid association preferences by performing a Delaunay triangulation of the lipid centres of mass, including the effects of the periodic boundary. The diagrams in figure 4 illustrate the calculation by drawing the (Voronoi) mesh, in which each line separating lipid tiles represents an association or adjacency between two lipids and an edge in the corresponding Delaunay triangulation. To characterize the lipid-lipid association preferences, we counted the number of lipid-lipid associations of each type, for example an edge between $PIP_2 - DOPS$, and compared these to the expected number for a random triangulation of a bilayer with an identical composition. The result provides a score relative to 1.0, which captures the extent to which that lipid association is favoured above (greater than 1.0) or below (less than 1.0) chance. We have also applied this calculation to groups of three lipids which we call a *triplet*, represented by a triangle on the Delaunay mesh.

We hypothesize that conditions which produce lipid pairs and triplets above chance will be more likely to create large and growing clusters in biological systems. The results are pictured in the electronic supplementary material, figure S14. We find that pure PPI triplets (for example, three adjacent $PI(4, 5)P_2$) are disfavoured relatively to chance, suggesting that their bulky headgroups inhibit direct adjacency. A comparison of the two 500 ns trajectories containing divalent cations shows that the presence of $Ca^{2+}$ correlates with heightened $PIP_2$ pairs, but not triplets, compared to $Mg^{2+}$, which occur below chance. Observations of adjacent $Ca^{2+}$-mediated $PIP_2$ pairs both above chance and in greater numbers than those observed with $Ca^{2+}$ suggests that these structures are more energetically favourable.

# 4. Discussion

This study considers the chemical and physical characteristics of asymmetric membranes containing the rare, yet biologically crucial, phosphoinositide class of lipids. Although cation-induced cluster formation has been observed *in vitro* and *in vivo* [7,18–28], the physical basis and mechanism of clustering has remained unclear on an atomic level. What are the molecular driving forces and principles of clustering and how are they affected by the chemical context when clusters are forming? The answer to these questions can suggest mechanisms for determining the size and prevalence of plasma membrane clusters rich in $PIP_2$, how they are regulated by, and how they regulate cell signalling on the membrane interface. We used a set of 15 simulations to independently vary the ionic conditions, net charge of the membrane, cholesterol content, leaflet composition asymmetry, and number and position of the phosphate groups on the inositol ring (PI, $PI(3, 5)P_2$, or $PI(4, 5)P_2$). We hypothesized that the close association driven by electrostatics and hydration status of divalent cations—particularly $Ca^{2+}$—create a 'molecular glue' capable of driving the self-association of PPIs.

Over the course of the simulations, we observe condensation of monovalent and divalent counterions on the negatively charged surface of the asymmetric membranes (figure 3). This effect has been observed experimentally with $PI(4, 5)P_2$-containing micelles [63], supported monolayers, [62,64] liposomes [27], large unilamellar vesicles and giant unilamellar vesicles [59], and cells [7]. Coarse-grained simulations of monolayers containing $PI(4, 5)P_2$ [3] and atomistic simulations of DOPC/DOPS membranes [65] have also shown $Ca^{2+}$ to lead to clustering. These results implicate the identity of the counterion in the formation of nano-clusters. Although $Ca^{2+}$ and $Mg^{2+}$ carry the same charge, they evoke distinctly different effects on charged bilayers. Hence, many of these differences must be attributed to their size and hydration free energy, which in turn generate different bonding patterns.

## 4.1. Charging

We find $Ca^{2+}$ is most strongly associated with the membrane, followed by $Na^+$, $K^+$, and then $Mg^{2+}$ (see the electronic supplementary material, figure S13). $Ca^{2+}$ in particular, is able to simultaneously bind to one, two, or three lipids, and of those connections, roughly half are interactions with $PIP_2$, whereas $Mg^{2+}$ predominantly binds just one lipid at a time, with $PIP_2$ in the minority. Monovalent ions fail to generate comparable charge density at close distances. $K^+$ forms far fewer bonds than $Na^+$, and both are typically limited to binding a single lipid. $Ca^{2+}$ binds PI less than $PIP_2$, and forms fewer 3-bridges.

The charge brought to the membrane surface by $Ca^{2+}$ is $+400e$, representing significant overcharging above the net $-300e$ charge carried by the lipids. This overcharging leads to the formation of a double layer containing the co-ion $Cl^-$. It is notable that the $Cl^-$ concentration in the double layer decays much more gradually when $Mg^{2+}$ is present compared to $Ca^{2+}$ (results not pictured here). When the divalent cations bind to the phosphate groups of PPIs, $Mg^{2+}$ binds closer to PPI phosphate groups than $Ca^{2+}$ but

much more $Ca^{2+}$ is present (figure 6). Simulations containing cholesterol have approximately 20% more bound $Ca^{2+}$, however, these simulations also have an asymmetric composition in this study.

This charging effect, in tandem with the formation of cation-mediated 3-bridges, represents the driving force for much of the resulting redistribution of the lipids in this bilayer. The slow accumulation of $Ca^{2+}$-driven 3-bridges, at the expense of 1- and 2-bridges, suggests the bilayer continues to undergo rearrangements as these nano-clusters develop. Coarse-grained models which account for both the condensed (Stern) and diffuse layers of cations at the lipid surface indicate that there is a cooperative effect in which cation binding is more favourable when lipids are clustered rather than dispersed [66]. These models suggest that the heightened $N$-bridges and resulting growth of lipid-lipid bond networks may proceed via feedback in which cations bind and enrich $PIP_2$, which attracts further cations as it forms larger or denser clusters.

## 4.2. Hydration

The strength of ion-lipid interactions is determined by the relationship between hydration and coordination of the ion in bulk coupled with the general geometry constraints from the membrane and the specific partial desolvation of the ion by $PIP_2$. Because $Ca^{2+}$ and $Mg^{2+}$ have different preferred coordination in bulk (with $Mg^{2+}$ ranging $6-8$ and $Ca^{2+}$ preferring octahedral 6 coordination) and corresponding hydration free energies, we expect this to influence their ability to incorporate a $PIP_2$ phosphate group in their first solvation shell. We find $Mg^{2+}$ retains more water, with more ordering in its first hydration shell (see the electronic supplementary material, figure S1). The high hydration free energy of $Mg^{2+}$ underlies its short-lived binding (see the electronic supplementary material, video SV2).

## 4.3. Ions drive rearrangements

Despite condensing on the phosphate groups of $PIP_2$, divalent cations do not significantly disrupt endogenous hydrogen bonds between lipids. It is not surprising that $Ca^{2+}$ directly mediates a very large number of intermolecular salt bridges between headgroup atoms which combine to form $N$-bridges. The absence of large compensatory changes to lipid-lipid hydrogen bonds is evidence that $Ca^{2+}$ acts as a 'molecular glue' by forming $O-Ca^{2+}-O$ salt bridges at the level of the lipid headgroups. We attribute the differences in cluster formation, diffusion, and lipid area across conditions to ion-mediated lipid interactions. That the cation binding effect does not interfere with lipid-hydrogen bonding implies that these bonds are available for manipulation by other components, for example lysine-containing peptides, which may use them to stabilize further clustering when they are present [67].

## 4.4. Lipid-lipid interactions

Intermolecular bonds do not convey the full scope of the attraction and repulsion between two lipids. Understanding those forces is necessary to predict the lateral distribution of lipids. One possible proxy for those attractive forces is the statistical likelihood that two lipids will occupy adjacent spaces in the bilayer. These likelihoods may reflect lipid-lipid association preferences if they are well-sampled either by lengthy simulations or sampling many lipids. We have studied these associations via Delaunay triangulation described in the Results section and identified some notable patterns in the results.

If we restrict our attention to the longest-running simulations, we find that almost all pair association probabilities show decreased preferences for adjacent $PIP_2$. This is consistent with steric repulsion owing to the bulky headgroup (see the electronic supplementary material, figure S14). By contrast, DOPE has a smaller headgroup and shows heightened associations with $PIP_2$, and extra opportunities for hydrogen bonding, reflected in the electronic supplementary material, figure S6 may also facilitate these interactions. While these pair association probabilities have an intuitive explanation, we surprisingly find that the specific combination of PI(4, 5)$P_2$ and $Ca^{2+}$ shows $PIP_2$ associations above simulations with $Mg^{2+}$. These probabilities also exceed the random probabilities of observing adjacent $PIP_2$. To demonstrate a causal relation between the $N$-bridges or lipid-lipid network and the propensity for $PIP_2$ to form enriched clusters, these heightened associations must be confirmed with free energy calculations or extremely long trajectories. They nevertheless provide a plausible connection to experiments.

Analysis of these likelihoods does not include additional interactions that could drive clustering. Snapshots in the electronic supplementary material, figure S5 indicate that $Ca^{2+}$ ions can enable $PIP_2$ to 'reach over' nearby lipids to form associations with next-nearest neighbours. That this binding is less prevalent in simulations with $Mg^{2+}$ suggests yet another specific feature of $Ca^{2+}$ binding that might influence the lateral distribution of the $PIP_2$. A more complete characterization of $PIP_2$ would include these next-nearest-neighbour interactions, explain the heightened association between otherwise bulky lipids, and account for the timescale by which these clusters form *in vivo*. We note that while cholesterol was included to mimic the physiological composition, the systematic effect of cholesterol was not studied in this work; we refer to a recent related study that investigates the effect of cholesterol on protein binding to $PIP_2$-containing bilayers [68].

## 4.5. Clusters

The implications of $PIP_2$ cluster growth depend on the precise conditions under which they form. Han *et al.* [57] have studied the structure of cation-stabilized cluster formation in a slightly different context, namely monolayers composed exclusively of $PIP_2$. They report a similar increase in area compressibility in the presence of $Ca^{2+}$ and find mixtures of ions may further stabilize clusters. They characterize clusters by direct lipid-lipid bonds, providing a more natural definition for a $PIP_2$-rich microenvironment and revealing linear, string-like clusters. Our results reported here for ion-mediated $PIP_2$ clusters are in qualitative agreement with Han *et al.* despite the difference in lipid microenvironment (their monolayers versus our bilayers, our inclusion of other lipids, and our significantly lower $PIP_2$ concentration). In either interpretation, the nanoclusters are viewed as assemblies generated by connected graphs (i.e. lipids act as nodes linked together either directly or through ions), rather than 2D domains associated with microphase separation.

## 4.6. Mechanical/structural properties

In order to relate highly-local molecular rearrangements to larger-length scale changes in the structure and dynamics of the bilayer, we computed representative macroscopic properties, namely diffusion coefficients and areas per lipid. $Ca^{2+}$ slows the diffusion of all lipids in our simulations (see the electronic supplementary material, figure S10) despite the fact that it coordinates the most with $PIP_2$ which represent only 10% of the inner leaflet. The area per lipid, however, is a more illuminating descriptor of the lipid packing. It reflects the denominator of the local charge density at a fixed lipid composition and also serves as an important order parameter for describing phase changes in lipid bilayers because it couples to the bilayer thickness and surface pressure.

In the presence of $Ca^{2+}$, we find a small but significant contraction of roughly 3% in both the projected bilayer areas and in the leaflet areas measured on a 3D mesh. In the small simulations produced for this study, this area contraction is strong enough to condense the entire bilayer even when the $PIP_2$ only populates the inner leaflet. In larger bilayers, this contraction might be less frustrated by periodic boundaries, and hence more likely to create spontaneous curvature. The area contraction owing to $Ca^{2+}$ is roughly equal and opposite to the area asymmetry generated by $PIP_2$ with phosphorylation at positions 4, 5 and 3, 5 compared to PI, which lacks these extra groups.

These results, taken together with the observation that $PIP_2$ has the slowest diffusion coefficient of all the membrane constituents we tested, even more so in the presence of $Ca^{2+}$, and that $Ca^{2+}$ can bridge multiple $PIP_2$ molecules, provide a potential mechanism to explain experimental $Ca^{2+}$-induced $PIP_2$ clusters. Specifically, enrichment of $PIP_2$ in a particular region that grows from 2- and 3-bridges to the nanometre scale, can lead to an area and fluidity mismatch between the two leaflets that is dissipated through the formation of curvature. This is consistent with bulges found on supported lipid monolayers containing PI(4, 5)$P_2$ in the presence of micromolar $Ca^{2+}$ imaged with atomic force microscopy [59] and shape instabilities on giant unilamellar vesicles containing PI(4, 5)$P_2$ exposed to excess $Ca^{2+}$ on the lipid exterior [69].

It is notable that the additional area owing to poly-phosphorylated inositol rings is consistent across ionic conditions in our simulations. This means that it is the solvent-facing phosphate groups, and not the $Ca^{2+}$, that provide the excess leaflet area. Hence any bulging area owing to $Ca^{2+}$ in experiments could be owing to its ability to locally enrich PPIs to a higher concentration compared to other cations. Second, our simulations might likewise predict that $Ca^{2+}$ bound to a leaflet containing a negative charge can generate an area contraction that creates invaginations, as long as the pre-existing $PIP_2$ concentration is high enough. We estimate that at 10% $PIP_2$, an area contraction of 3% across leaflets with centres of mass

separated by 3 nm would correspond to the spontaneous curvature of a 235 nm vesicle (positive if the area contraction is on the outside). The modulation of lipid area by $Ca^{2+}$ might contribute to $PIP_2$ phase separation directly, or facilitate variation in spontaneous curvature, which causes lipid sorting. In either case, the specific features of headgroup-ion interactions identified in these simulations can plausibly play a role in sequestering $PIP_2$ into highly enriched microdomains.

# 5. Conclusion

In conclusion, we have used advances in molecular dynamics to investigate the molecular mechanisms driving lateral organization of phosphoinositides in asymmetric bilayers. A wealth of experimental data is available showing interactions of phosphoinositdes with cytoskeletal-binding proteins, and a natural extension of this work will be to elucidate how the lipid bilayer couples to the protein framework immediately beneath it in the cell membrane. Together, the molecular interactions at the membrane-cytoskeletal interface can produce an integrated model for how these fascinating lipids perform their many cellular functions.

Data accessibility. The protocols are available as part of the electronic supplementary material. The codes and scripts along with complete documentation are available as part of the biophyscode software developed by the authors. The distribution of biophyscode is through an open-source license through GitHub and can be accessed using the link: https://biophyscode.github.io.

Authors' contributions. R.P.B. and D.S. performed the molecular simulations, R.B.P., D.S., P.A.G., and R.R., performed all the analysis. All authors contributed to the discussion and the writing of the manuscript. R.P.B. and D.S. contributed to the work equally.

Competing interests. There are no conflicts to declare.

Funding. This work was supported in part by grant nos. NIH GM111942 and GM136259 (P.A.G. and R.R.), NIH CA193417 (R.R.) and NSF DMR1120901 (R.R.). Computational resources were provided in part by the grant no. MCB060006 from XSEDE, by Penn MRSEC, and through Penn Institute for Computational Science (PICS).

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
