## [Reviewer comments · Royal Society Open Science]

Review History

RSOS-192208.R0 (Original submission)

Review form: Reviewer 1

Is the manuscript scientifically sound in its present form?

Yes

Are the interpretations and conclusions justified by the results?

Yes

Is the language acceptable?

Yes

Do you have any ethical concerns with this paper?

No

Have you any concerns about statistical analyses in this paper?

No

Recommendation?

Major revision is needed (please make suggestions in comments)

Comments to the Author(s)

This manuscript investigates the ion-lipid interactions between model membranes that contain PIP2 lipids. This is a well-designed study and has many insightful results based on detailed analysis of their simulations. Overall, I find this work worthy of publishing but my comments/suggestions below should be addressed.

General Comments:

1. Ion force field: The authors note in the methods section that past work has suggested that the Ca²⁺ force field might over bind to anions. It is not expected here that the author tackle refitting Ca²⁺ parameters, but should be more upfront in the discussion/conclusions section that the results might be biased to potentially over binding of calcium to anionic lipids. For Na⁺, I am assuming the authors used the modified NBFIX form for this force field published in: <https://pubs.acs.org/doi/10.1021/jp401512z>. Please confirm.
2. Convergence Test: At the end of the methods section there is a statement that convergence was assessed based on Figures S1 and S2. This is not clear and figure S2 is only for the two longer simulations of 500ns. What about all the other simulations that were 100ns? Convergence in the RDF and surface area of the membrane vs. time should be provided to verify that structurally the bilayers in these short runs are stabilized. The subtle ion-membrane interaction can sometimes require longer times than expected.
3. Calcium binding (not converged): The authors state on page 4 (lines 50-55) that the charging of the bilayer is slow and based on figures like that of Figure 3 suggest 500ns may not be enough to fully probe binding and at best there is only 50ns of equilibrated data. The authors do note that structurally the systems might require more time to form clusters but clearly Figure 3 might suggest that even for ion binding more time might be needed. An honest assessment of this should be provided in the discussion or extending the simulations longer.
4. Diffusion: The ~20% reduction in diffusion for the PIP2 lipids is not as low as I would expect for the formation of large clusters. How was the diffusion calculated? Is the bilayer leaflet drift ignored or included in this?

Specific Comment

Page 2 (column 2: last line): change "timescale of 500-100ns" to "timescale of 500-1000ns"

Review form: Reviewer 2

Is the manuscript scientifically sound in its present form?

Yes

Are the interpretations and conclusions justified by the results?

Yes

Is the language acceptable?

Yes

Do you have any ethical concerns with this paper?

No

Have you any concerns about statistical analyses in this paper?

No

Recommendation?

Accept with minor revision (please list in comments)

Comments to the Author(s)

This computational study uses atomistic simulations of complex asymmetric bilayers to study the effects of cation interaction on bilayer phosphoinositides. This is an important and interesting problem of likely broad interest, as both these lipids and ions, and therefore their interactions, have enormous physiological significance. The major findings are that calcium ions effectively desolvate to induce networks/clusters of PPIs, clustering up to 3 lipids per ion. This effect is highly specific for a particular lipid-ion pair (PI45P2+Ca), as neither PI35P2-Ca nor PI45P2-Mg show the same effects. These insights are intriguing because of the unique physiological importance of both Ca and PI45P2, although the manuscript does not go much beyond implication in this context. The general trends of these results are also well supported by both experimental observations and by quantum mechanical modeling.

Overall, I found this work to be a solid and thoughtful contribution to a well-developed topic. I have no major concerns beyond the fact that most of the result were rather predictable from the rich recent literature on this subject. Nevertheless, I think this manuscript is a useful contribution.

Beyond that, I have a few minor points that I think are worth addressing:

1. Han et al seem to have come to similar conclusions, though apparently studying quite different systems (doi: 10.1021/acs.jpcc.9b10951). Thus, it seems to me that these stories are complementary rather than competing, but their results and approaches should be thoroughly compared and contrasted with those in this manuscript.
2. The correspondence to experimental measurements could use more emphasis. The general trends seem consistent, but are there some specific molecular-level measurements beyond microscopic clustering or monolayers that could be compared to the simulation results? In some aspects, there are major mis-fits to experiments: namely, in the Ca-induced condensation of the PIP2. According to Fig9, Ca condenses PIP2 by about 5%, which is dramatically lower than what has been reported in monolayer experiments. This divergence should be discussed.
3. Semantic quibble, but I do not think "physiologically composed" accurately describes a model that is likely very far from any real mammalian PM.
4. It has recently become clear that constructing asymmetric membranes properly is not a trivial matter (doi: 10.1016/j.bpj.2018.12.016). There are likely to be imbalances between the leaflets which could build up stress in the bilayer and necessarily give results that are highly dependent on the initial configurations. Asymmetry per se is not the focus of this manuscript, so this is not a fatal flaw, but I believe the authors should address this important caveat.
5. I don't understand the "pseudocluster" terminology. As far as I'm aware, 'cluster' is not a highly specified term. These stable aggregates of lipids and ions are clearly clusters. Perhaps this is to avoid the impression that these are the equilibrium clusters, since they are limited by the size of the box and timescale of the simulation. But I'm not sure that "pseudocluster" does this job well.
6. I liked the critical discussion of force fields and their potential influence in the Methods. Perhaps it would be worthwhile to recapitulate in the Discussion, with particular focus on the possible influence (or lack thereof) of force field parameters on the key results.
7. I find Fig7 to be an interesting result, but confusingly presented. I don't know the C36 convention which makes it very difficult to interpret these figures.
8. Fig5C is mislabeled 'Ca'
9. I think the last experimental finding is quite important. If I understand correctly, it suggests that experimentally observed clusters should be thought of as assemblies of string-like objects; rather like webs than 2D domains. This is a surprising finding and I think one worth emphasizing.

10. The inclusion of cholesterol and asymmetry is highlighted as lending physiological relevance, which is fine and correct, but how these factors influence the central observations is hardly discussed. Perhaps there are not notable effects, but that in itself is worth noting, especially in the context of other recent results that have reported PIP2-Ca interactions in different (simpler) membranes.

Decision letter (RSOS-192208.R0)

20-Feb-2020

Dear Dr Radhakrishnan,

The editors assigned to your paper ("Divalent cations bind to phosphoinositides to induce ion and isomer specific propensities for nano-cluster initiation in bilayer membranes") have now received comments from reviewers. We would like you to revise your paper in accordance with the referee and Associate Editor suggestions which can be found below (not including confidential reports to the Editor). Please note this decision does not guarantee eventual acceptance.

Please submit a copy of your revised paper before 14-Mar-2020. Please note that the revision deadline will expire at 00.00am on this date. If we do not hear from you within this time then it will be assumed that the paper has been withdrawn. In exceptional circumstances, extensions may be possible if agreed with the Editorial Office in advance. We do not allow multiple rounds of revision so we urge you to make every effort to fully address all of the comments at this stage. If deemed necessary by the Editors, your manuscript will be sent back to one or more of the original reviewers for assessment. If the original reviewers are not available, we may invite new reviewers.

- Data accessibility

It is a condition of publication that all supporting data are made available either as supplementary information or preferably in a suitable permanent repository. The data accessibility section should state where the article's supporting data can be accessed. This section

should also include details, where possible of where to access other relevant research materials such as statistical tools, protocols, software etc can be accessed. If the data have been deposited in an external repository this section should list the database, accession number and link to the DOI for all data from the article that have been made publicly available. Data sets that have been deposited in an external repository and have a DOI should also be appropriately cited in the manuscript and included in the reference list.

If you wish to submit your supporting data or code to Dryad (<http://datadryad.org/>), or modify your current submission to dryad, please use the following link:
<http://datadryad.org/submit?journalID=RSOS&manu=RSOS-192208>

- **Competing interests**

- **Authors' contributions**

- **Acknowledgements**

- **Funding statement**

Kind regards,

Andrew Dunn

on behalf of Professor Diwakar Shukla (Associate Editor) and Pietro Cicuta (Subject Editor)
openscience@royalsociety.org

Comments to Author:

Reviewers' Comments to Author:

Reviewer: 1

Comments to the Author(s)

This manuscript investigates the ion-lipid interactions between model membranes that contain PIP2 lipids. This is a well-designed study and has many insightful results based on detailed analysis of their simulations. Overall, I find this work worthy of publishing but my comments/suggestions below should be addressed.

General Comments:

1. Ion force field: The authors note in the methods section that past work has suggested that the Ca²⁺ force field might over bind to anions. It is not expected here that the author tackle refitting Ca²⁺ parameters, but should be more upfront in the discussion/conclusions section that the results might be biased to potentially over binding of calcium to anionic lipids. For Na⁺, I am assuming the authors used the modified NBFIX form for this force field published in: <https://pubs.acs.org/doi/10.1021/jp401512z>. Please confirm.

2. Convergence Test: At the end of the methods section there is a statement that convergence was assessed based on Figures S1 and S2. This is not clear and figure S2 is only for the two longer simulations of 500ns. What about all the other simulations that were 100ns? Convergence in the RDF and surface area of the membrane vs. time should be provided to verify that structurally the bilayers in these short runs are stabilized. The subtle ion-membrane interaction can sometimes require longer times than expected.

3. Calcium binding (not converged): The authors state on page 4 (lines 50-55) that the charging of the bilayer is slow and based on figures like that of Figure 3 suggest 500ns may not be enough to fully probe binding and at best there is only 50ns of equilibrated data. The authors do note that structurally the systems might require more time to form clusters but clearly Figure 3 might suggest that even for ion binding more time might be needed. An honest assessment of this should be provided in the discussion or extending the simulations longer.

4. Diffusion: The ~20% reduction in diffusion for the PIP2 lipids is not as low as I would expect for the formation of large clusters. How was the diffusion calculated? Is the bilayer leaflet drift ignored or included in this?

Specific Comment

Page 2 (column 2: last line): change "timescale of 500-100ns" to "timescale of 500-1000ns"

Reviewer: 2

Comments to the Author(s)

This computational study uses atomistic simulations of complex asymmetric bilayers to study the effects of cation interaction on bilayer phosphoinositides. This is an important and interesting problem of likely broad interest, as both these lipids and ions, and therefore their interactions, have enormous physiological significance. The major findings are that calcium ions effectively desolvate to induce networks/clusters of PPIs, clustering up to 3 lipids per ion. This effect is highly specific for a particular lipid-ion pair (PI45P2+Ca), as neither PI35P2-Ca nor PI45P2-Mg show the same effects. These insights are intriguing because of the unique physiological importance of both Ca and PI45P2, although the manuscript does not go much beyond implication in this context. The general trends of these results are also well supported by both experimental observations and by quantum mechanical modeling.

Overall, I found this work to be a solid and thoughtful contribution to a well-developed topic. I have no major concerns beyond the fact that most of the result were rather predictable from the rich recent literature on this subject. Nevertheless, I think this manuscript is a useful contribution.

Beyond that, I have a few minor points that I think are worth addressing:

1. Han et al seem to have come to similar conclusions, though apparently studying quite different systems (doi: 10.1021/acs.jpcc.9b10951). Thus, it seems to me that these stories are complementary rather than competing, but their results and approaches should be thoroughly compared and contrasted with those in this manuscript.
2. The correspondence to experimental measurements could use more emphasis. The general trends seem consistent, but are there some specific molecular-level measurements beyond microscopic clustering or monolayers that could be compared to the simulation results? In some aspects, there are major mis-fits to experiments: namely, in the Ca-induced condensation of the PIP2. According to Fig9, Ca condenses PIP2 by about 5%, which is dramatically lower than what has been reported in monolayer experiments. This divergence should be discussed.
3. Semantic quibble, but I do not think “physiologically composed” accurately describes a model that is likely very far from any real mammalian PM.
4. It has recently become clear that constructing asymmetric membranes properly is not a trivial matter (doi: 10.1016/j.bpj.2018.12.016). There are likely to be imbalances between the leaflets which could build up stress in the bilayer and necessarily give results that are highly dependent on the initial configurations. Asymmetry per se is not the focus of this manuscript, so this is not a fatal flaw, but I believe the authors should address this important caveat.
5. I don't understand the “pseudocluster” terminology. As far as I'm aware, ‘cluster’ is not a highly specified term. These stable aggregates of lipids and ions are clearly clusters. Perhaps this is to avoid the impression that these are the equilibrium clusters, since they are limited by the size of the box and timescale of the simulation. But I'm not sure that “pseudocluster” does this job well.
6. I liked the critical discussion of force fields and their potential influence in the Methods. Perhaps it would be worthwhile to recapitulate in the Discussion, with particular focus on the possible influence (or lack thereof) of force field parameters on the key results.
7. I find Fig7 to be an interesting result, but confusingly presented. I don't know the C36 convention which makes it very difficult to interpret these figures.
8. Fig5C is mislabeled ‘Ca’
9. I think the last experimental finding is quite important. If I understand correctly, it suggests that experimentally observed clusters should be thought of as assemblies of string-like objects; rather like webs than 2D domains. This is a surprising finding and I think one worth emphasizing.
10. The inclusion of cholesterol and asymmetry is highlighted as lending physiological relevance, which is fine and correct, but how these factors influence the central observations is hardly discussed. Perhaps there are not notable effects, but that in itself is worth noting, especially in the context of other recent results that have reported PIP2-Ca interactions in different (simpler) membranes.

Author's Response to Decision Letter for (RSOS-192208.R0)

See Appendix A.

RSOS-192208.R1 (Revision)

Review form: Reviewer 1

Is the manuscript scientifically sound in its present form?

Yes

Are the interpretations and conclusions justified by the results?

Yes

Is the language acceptable?

Yes

Do you have any ethical concerns with this paper?

No

Have you any concerns about statistical analyses in this paper?

No

Recommendation?

Accept as is

Comments to the Author(s)

Changes made are sufficient

Review form: Reviewer 2

Is the manuscript scientifically sound in its present form?

Yes

Are the interpretations and conclusions justified by the results?

Yes

Is the language acceptable?

Yes

Do you have any ethical concerns with this paper?

No

Have you any concerns about statistical analyses in this paper?

No

Recommendation?

Accept as is

Comments to the Author(s)

The authors have done a commendable job of addressing all reviewer concerns. I believe this manuscript is ready for publication.

Decision letter (RSOS-192208.R1)

14-Apr-2020

Dear Dr Radhakrishnan,

It is a pleasure to accept your manuscript entitled "Divalent cations bind to phosphoinositides to induce ion and isomer specific propensities for nano-cluster initiation in bilayer membranes" in its current form for publication in Royal Society Open Science. The comments of the reviewer(s) who reviewed your manuscript are included at the foot of this letter.

Please note that we require all authors to have an active email address, but currently slochow@sas.upenn.edu is not receiving messages from us - please can you check with your colleague to ensure that either this address is able to receive messages or that we can get an alternative email address?

on behalf of Professor Diwakar Shukla (Associate Editor) and Pietro Cicuta (Subject Editor)
openscience@royalsociety.org

Reviewer comments to Author:

Reviewer: 1

Comments to the Author(s)
changes made are sufficient

Reviewer: 2

Comments to the Author(s)

The authors have done a commendable job of addressing all reviewer concerns. I believe this manuscript is ready for publication.

Appendix A

Reviewer: 1

1. This manuscript investigates the ion-lipid interactions between model membranes that contain PIP2 lipids. This is a well-designed study and has many insightful results based on detailed analysis of their simulations. Overall, I find this work worthy of publishing but my comments/suggestions below should be addressed.

Author Response: Thank You.

2. Ion force field: The authors note in the methods section that past work has suggested that the Ca²⁺ force field might over bind to anions. It is not expected here that the author tackle refitting Ca²⁺ parameters, but should be more upfront in the discussion/conclusions section that the results might be biased to potentially over binding of calcium to anionic lipids. For Na⁺, I am assuming the authors used the modified NBFIX form for this force field published in: <https://pubs.acs.org/doi/10.1021/jp401512z>. Please confirm.

Author Response: We have also clarified that there has been great effort in the community for continually improving the ion force fields, and in this work we have not explicitly attempted to refine force-field parameters surrounding ions. We confirm that we have indeed used the NBFIX parameters. This is also clarified in the main text.

Please also see our response to Reviewer 2, question 6 below for more notes regarding the force field.

3. Convergence Test: At the end of the methods section there is a statement that convergence was assessed based on Figures S1 and S2. This is not clear and figure S2 is only for the two longer simulations of 500ns. What about all the other simulations that were 100ns? Convergence in the RDF and surface area of the membrane vs. time should be provided to verify that structurally the bilayers in these short runs are stabilized. The subtle ion-membrane interaction can sometimes require longer times than expected.

Author Response: We have provided RDFs for other systems which ran for 100 ns. However, we have also noted that in this work, mainly the results focus on Ca and Mg simulations that have run for longer (500 ns). See Fig. S2.

4. The authors state on page 4 (lines 50-55) that the charging of the bilayer is slow and based on figures like that of Figure 3 suggest 500ns may not be enough to fully probe binding and at best there is only 50ns of equilibrated data. The authors do note that structurally the systems might require more time to form clusters but clearly Figure 3 might suggest that even for ion binding more time might be needed. An honest assessment of this should be provided in the discussion or extending the simulations longer.

Author Response: We thank the reviewer for this comment. We believe we have presented the results and interpretation in a direct fashion. What gives us confidence is the **reproducibility of the observations in our replicates**. We discuss this aspect in the revised text. We also note that as the clusters grow, PIP2 is also limiting as we are working

with a constant number of PIP2 molecules in the system in contrast to a situation in which the chemical potential of PIP2 is a constant. Both these aspects limit our ability to grow clusters even with longer simulations. However, we note the referee's point that it is not just about the clusters and that even though we see reproducible trends in our replicates with respect to ion binding, we cannot rule out a scenario which could show a rearrangement with respect to ion binding at a longer time frame.

5. Diffusion: The ~20% reduction in diffusion for the PIP2 lipids is not as low as I would expect for the formation of large clusters. How was the diffusion calculated? Is the bilayer leaflet drift ignored or included in this?

Author Response: Our bilayers have no net drift artifacts at least partly because we have used CHARMM "special" water which includes non-bonded interactions on the hydrogen atoms. This is the default for the CHARMM36 force field. Diffusion is computed from a linear fit of the lateral mean-squared displacement after accounting for periodic boundary conditions. Our simulations use a fixed 20% concentration of PIP2 and therefore do not achieve the local concentration of PIP2 which is characteristic of a mature or homogeneous PIP2 cluster. We agree with the reviewer that further enrichment of PIP2 would likely reduce its diffusion rate by enhancing the effects of the "molecular glue" however we have sampled a lower concentration in order to investigate the conditions that generate these clusters rather than the behavior of a pre-formed cluster.

6. Page 2 (column 2: last line): change "timescale of 500-100ns" to "timescale of 500-1000ns"

Author Response: We have made this change.

Reviewer: 2

This computational study uses atomistic simulations of complex asymmetric bilayers to study the effects of cation interaction on bilayer phosphoinositides. This is an important and interesting problem of likely broad interest, as both these lipids and ions, and therefore their interactions, have enormous physiological significance. The major findings are that calcium ions effectively desolvate to induce networks/clusters of PPIs, clustering up to 3 lipids per ion. This effect is highly specific for a particular lipid-ion pair (PI45P2+Ca), as neither PI35P2-Ca nor PI45P2-Mg show the same effects. These insights are intriguing because of the unique physiological importance of both Ca and PI45P2, although the manuscript does not go much beyond implication in this context. The general trends of these results are also well supported by both experimental observations and by quantum mechanical modeling.

Author Response: Thank You. The implications beyond the results stated get interesting when protein recruitment and subsequent cellular processes like cytoskeletal rearrangement are considered. We have added a few links in this direction based on recent literature by revising the introduction. See example [[10.1016/j.bbrc.2018.07.155](https://doi.org/10.1016/j.bbrc.2018.07.155)].

Overall, I found this work to be a solid and thoughtful contribution to a well-developed topic. I have no major concerns beyond the fact that most of the result were rather predictable from the rich recent literature on this subject. Nevertheless, I think this manuscript is a useful contribution.

Author Response: Thank You.

1. Han et al seem to have come to similar conclusions, though apparently studying quite different systems (doi: 10.1021/acs.jpcc.9b10951). Thus, it seems to me that these stories are complementary rather than competing, but their results and approaches should be thoroughly compared and contrasted with those in this manuscript.

Author Response: We have included the discussion of this paper in the revised manuscript. Han et al. characterized clusters in a somewhat similar fashion, albeit in monolayers composed entirely of PIP2. Our work builds on their recommendation to extend this investigation to (1) bilayers and (2) bilayers with other lipid species. Their analysis proceeds from a different definition of an “edge” in the lipid-association graph, namely the proximity between phosphate oxygen groups. Our definition of an “edge” is a lipid-cation-lipid bond, that is, a desolvated cation shared between two lipids. Both definitions are valid ways to describe the concept of a cluster, however we are unlikely to see the edges from Han et al. in our simulations because we have a much lower PIP2 concentration and direct PIP2 interactions are statistically much less likely because there are many other lipids. This difference enables us to investigate the statistical likelihood of forming new PIP2 pairs (see the final part of the results section). The Han et al. analysis provides a characterization of the structure of a mature 100% PIP2 cluster, with the same underlying molecular basis as the clusters described in our study. Much longer simulations of local PIP2 enrichment would bridge the gap between our respective conditions. We contend that including direct cation interactions helps inform a model for generating clusters, whereas the topology of direct lipid-lipid bonds in their method, may be more relevant for the properties of an enriched, stabilized cluster. We have added a discussion in the main text.

2. The correspondence to experimental measurements could use more emphasis. The general trends seem consistent, but are there some specific molecular-level measurements beyond microscopic clustering or monolayers that could be compared to the simulation results? In some respects, there are major mis-fits to experiments: namely, in the Ca-induced condensation of the PIP2. According to Fig9, Ca condenses PIP2 by about 5%, which is dramatically lower than what has been reported in monolayer experiments. This divergence should be discussed.

Author Response: We have identified three broad classes of simulation and experimental validation: (1) the qualitative presence of lipid associations which might reflect cluster formation, (2) observations which are consistent with the desolvation energies reported by QM/MM simulations, and (3) collective changes in bilayer mechanical properties. We believe that the increased area compressibility, lower projected areas and leaflet areas, and lowered diffusion presented in this work are the best possible matches to experiments. The

Ca²⁺-induced area condensation in particular influences many observable bilayer physical properties, namely the lipid order and the lateral pressure profile. As the reviewer notes, we have reported a general agreement with monolayer experiments.

However, the effects of the area condensation observed in experiments may appear to be larger than what we report in our simulations. We have cited the experiments of the Janmey group [10.1021/bi9007879] in discussing the experimental observations of cation induced area condensation. The reports in this paper suggests that the effect in monolayer is a reduction of 10-40% reduction of area.

There are a few reasons limiting our ability to draw further conclusions from our work or seek a closer fit to monolayer experiments. (a) First, the pressure-area isotherms gleaned from monolayer experiments represent the ground truth against which a lipid force field is measured. Insofar as experiments show stronger area condensation, this may reflect the approximate nature of the models, we cannot rule out force-field artifacts. We have added this point to the discussion of the force field. (b) Second, the bilayer system is not guaranteed to match the exact lateral pressure profile of a monolayer. Instead, bilayer models can only reproduce the area per lipid, and even then, they may do so more effectively under particular ensembles. The diversity of lipids and cations in this study may require further parameterization to match not only the surface area isotherms from monolayer experiments, but also the changes they experience under high ionic strength. (c) Lastly, we expect that periodic boundary conditions have a strong influence on this area condensation effect, since they affect the undulations and hence the net curvature. We have noted this caveat in the discussion.

3. Semantic quibble, but I do not think “physiologically composed” accurately describes a model that is likely very far from any real mammalian PM.

Author Response: We have changed the phrase physiologically-composed to asymmetric.

4. It has recently become clear that constructing asymmetric membranes properly is not a trivial matter (doi: 10.1016/j.bpj.2018.12.016). There are likely to be imbalances between the leaflets which could build up stress in the bilayer and necessarily give results that are highly dependent on the initial configurations. Asymmetry per se is not the focus of this manuscript, so this is not a fatal flaw, but I believe the authors should address this important caveat.

Author Response: We have made clear that in preparing the asymmetric leaflets, we chose to focus on the compositions and balanced the number of lipids on each leaflet based on area per head group and the total leaflet area. These effects are in part mediated by our use of semi-isotropic pressure coupling to produce an approximately tensionless bilayer. In general, it is quite difficult to ensure the differential stress build-up across both the leaflets in an asymmetric simulation. As the reviewer has noted, we have not sought to eliminate the imbalance in the lateral pressure profile caused by the excess area imparted by PIP₂. Recent work [10.1021/acs.jctc.5b00232] has indicated that area asymmetries below 5% have negligible effects on bilayer properties (e.g. thickness, diffusion, and order

parameters). Our area mismatch is well below this 5% threshold. We would expect that the asymmetry would have far more significant implications in a bilayer further enriched in the PIP2 thanks to its bulky headgroups. While there is a large body of literature that considers the consequences of general lipid asymmetry on bilayer stress and properties, the question of how the polyphosphoinositides affect these stresses is worthy of further study.

5. I don't understand the "pseudocluster" terminology. As far as I'm aware, 'cluster' is not a highly specified term. These stable aggregates of lipids and ions are clearly clusters. Perhaps this is to avoid the impression that these are the equilibrium clusters, since they are limited by the size of the box and timescale of the simulation. But I'm not sure that "pseudocluster" does this job well.

Author Response: We have replaced pseudocluster with cluster and noted the caveat that the cluster size may be limited by the size of the box, timescale of the simulation, and the composition of the lipids (or the fact that the system holds the lipid numbers constant).

6. I liked the critical discussion of force fields and their potential influence in the Methods. Perhaps it would be worthwhile to recapitulate in the Discussion, with particular focus on the possible influence (or lack thereof) of force field parameters on the key results.

Author Response:

In response to both reviewers (see also reviewer 1, question 2) we have added text to the discussion to clarify that these findings are contingent on force field accuracy, particularly with regard to ion interactions. We contend that the qualitative match to experiments which characterize PIP2 clusters provides evidence that the altered lipid interactions due to calcium ions are not an artifact of the simulation. The following was added to the main text:

The chemical and physical implications of divalent cation binding in this study are subject to the accuracy of the underlying force field, and in particular, the precise strength of cation-ion interactions with PIP2 phosphate groups. We expect that the strength of these interactions affects both the lipid-lipid associations configurations as well as the altered mechanical properties. Further parameterization of the PIP2-ion interactions may improve our estimates of binding affinity and thereby facilitate predictions for the precise stoichiometry for stable physiological PIP2 nanoclusters. Recent comparisons of ad hoc adjustments of force field parameters to correct ion overbinding indicate that both the nonbonded fix (NBFIX) and electronic continuum correction (ECC) may improve the accuracy of ion, however these depend somewhat on the precise application [10.1021/acs.jpcc.7b12510, 10.1021/acs.jctc.9b00813].

7. I find Fig7 to be an interesting result, but confusingly presented. I don't know the C36 convention which makes it very difficult to interpret these figures.

Author Response: We have improved the figure 7 caption to explain the CHARMM36 nomenclature. Figure 7 provides a compact map of the oxygen atoms which are the most

likely to participate in a bridge. The axes highlight the phosphate oxygens at the 1, 4, and 5 positions in bold. The intensity is normalized across simulations by the color bars in order to emphasize that Ca^{2+} participates in more bridges than the other cations. Within each panel, the intensity indicates the oxygen atoms most likely to participate in a bridge. Han et al. (see response to question 3, Reviewer 2 above), have performed a similar decomposition of their edges to see which atoms associate the most, finding that the non-protonated 5-phosphate oxygens form the most edges. Our simulations have far more cation-mediated bridges between the 5- and 1-phosphate positions (diester phosphate, with oxygens prefixed with OP1) than Han et al., perhaps due to the fact that their 100% PIP2 composition alters the packing of these lipids by restricting the motion of the flexible headgroups. We have indicated this in the text along with the discussion of Han et al outlined above.

Addition to the Figure 7 caption: Atom names follow the CHARMM36 convention in which the OP4 prefix indicates the 4-phosphate oxygens. Our PIP2 are protonated at the OP52 position where there is a corresponding absence of bridges (lower intensity above). The prefix OP1 indicates the diester phosphate.

8. Fig5C is mislabeled 'Ca'

Author Response: Thank you for spotting this error. We have corrected the figure.

9. I think the last experimental finding is quite important. If I understand correctly, it suggests that experimentally observed clusters should be thought of as assemblies of string-like objects; rather like webs than 2D domains. This is a surprising finding and I think one worth emphasizing.

Author Response: We agree and have added a comment on this in the discussion.

10. The inclusion of cholesterol and asymmetry is highlighted as lending physiological relevance, which is fine and correct, but how these factors influence the central observations is hardly discussed. Perhaps there are not notable effects, but that in itself is worth noting, especially in the context of other recent results that have reported PIP2-Ca interactions in different (simpler) membranes.

Author Response: We agree that the influence of cholesterol is extremely important to questions of lipid structure and packing, and these in turn are important to our results. Questions 1, 2, and 4 from Reviewer 2, have each at least obliquely raised the question of whether an alternate bilayer structure (monolayer or asymmetric bilayer) affects the underlying results. Cholesterol adds a third dimension to these questions. We have considered the effects of cholesterol in a recent work that includes protein effects [10.1074/jbc.RA118.005552], however this question in the context of the current manuscript requires more controls and quantification of lipid packing before we can draw firm conclusions.